# Understanding the Complexity of Sjögren’s Syndrome: Remarkable Progress in Elucidating NF-κB Mechanisms

**DOI:** 10.3390/jcm9092821

**Published:** 2020-08-31

**Authors:** Margherita Sisto, Domenico Ribatti, Sabrina Lisi

**Affiliations:** Department of Basic Medical Sciences, Neurosciences and Sensory Organs (SMBNOS), Section of Human Anatomy and Histology, University of Bari “Aldo Moro”, 70124 Bari, Italy; domenico.ribatti@uniba.it (D.R.); sabrina.lisi@uniba.it (S.L.)

**Keywords:** Sjögren’s syndrome, NF-κB, inflammation

## Abstract

Sjögren’s syndrome (SS) is a systemic autoimmune inflammatory disease with a poorly defined aetiology, which targets exocrine glands (particularly salivary and lachrymal glands), affecting the secretory function. Patients suffering from SS exhibit persistent xerostomia and keratoconjunctivitis sicca. It is now widely acknowledged that a chronic grade of inflammation plays a central role in the initiation, progression, and development of SS. Consistent with its key role in organizing inflammatory responses, numerous recent studies have shown involvement of the transcription factor nuclear factor κ (kappa)-light-chain-enhancer of activated B cells (NF-κB) in the development of this disease. Therefore, chronic inflammation is considered as a critical factor in the disease aetiology, offering hope for the development of new drugs for treatment. The purpose of this review is to describe the current knowledge about the NF-κB-mediated molecular events implicated in the pathogenesis of SS.

## 1. Introduction

The nuclear factor κ (kappa)-light-chain-enhancer of activated B cells (NF-κB) is a pleiotropic regulator of many cellular signalling pathways activated in response to a wide variety of stimuli linked to inflammation. Once activated, this B cell enhancer plays an important role in the pathogenesis of several inflammatory autoimmune diseases, including Sjögren’s syndrome (SS) [1]. SS presents lymphocytic infiltration of the salivary glands (SGs) and lachrymal glands as the characteristic hallmark resulting in chronic inflammation. A dry mouth and dry eyes, resulting in keratoconjunctivitis sicca and xerostomia, are common complaints in SS [2]. NF-κB is a family of DNA-binding proteins that regulates many cellular processes, notably the immune response and inflammation, influencing the transcription of a broad array of pro-inflammatory cytokines [3]. NF-κB is ubiquitously expressed in SGs, and the constitutive NF-κB activation observed in primary SS (pSS) is associated with NF-κB release and nuclear translocation of NF-κB, to focal infiltrated lymphocytes and the acinar epithelium of patients with pSS, to regulate the pro-inflammatory gene transcriptions [4]. However, the role of NF-κB in pSS remains to be clarified in detail. This article provides an update on the current state of knowledge about the relationship between NF-κB-molecular pathway activation in SGs and the chronic inflammation characterizing pSS, with the aim of providing a strong basis for a better understanding of the signal transduction pathways mediating the induction of NF-κB in pSS SGs, in order to allow this disease to be manipulated, to gain therapeutic benefit.

## 2. Sjögren’s Syndrome

The chronic inflammatory autoimmune disorder SS arises as primary SS (pSS) and, when linked with another underlying systemic autoimmune disorder, such as scleroderma, systemic lupus erythematosus (SLE), or rheumatoid arthritis (RA), is defined as secondary SS [5]. The evolution to non-Hodgkin’s lymphoma occurs in a larger percentage of SS patients than in the normal population [6,7]. The clinical hallmarks of SS, keratoconjunctivitis sicca and xerostomia [2], can be confirmed by various objective tests highlighting significant functional impairment of the SGs and lachrymal glands [8]. The involvement of these glands is characterized by focal infiltrating lymphocytes that surround the ducts and, in some patients, extend and replace the secretory functional units. Although infiltration of the SGs by lymphocytes is a hallmark of SS [9,10], multiple cytokines are upregulated, even in the absence of lymphocytic infiltrates, and have a direct effect on SGs epithelial cells (SGEC). Interestingly, substantial new evidence supports the role of epithelia in the production of constitutive or inducible mediators of the innate and acquired immune responses. The picture that emerges shows intrinsically activated SGEC that induce and promote chronic inflammatory reactions [11]. For this reason, on the basis of clinical observation, pSS was defined as an ‘‘autoimmune epithelitis’’ [12]. Indeed, SGEC are capable of releasing many cytokines that result overexpressed and thus act as key molecules in chronic inflammation, contributing to both systemic and exocrine manifestations of pSS [13,14,15,16,17,18,19,20,21,22,23,24,25,26,27,28,29]. A number of explanations has been offered for the dysregulated cytokine network in pSS and, in the past, the presence of anti-Ro/SSA and anti-La/SSB antibodies was shown to be related to increased glandular and extra-glandular manifestations. Important findings provided evidence for a pathogenic role of autoantibodies, demonstrating that anti-Ro/SSA autoantibodies stimulate the production of pro-inflammatory cytokines such as IL-6 and IL-8 by human SGEC from healthy donors, promoting chronic inflammatory reactions [25]. Furthermore, pSS autoantibodies can promote the activation of the NF-κB pathway, leading to the overexpression of multiple proangiogenic/pro-inflammatory factors. Indeed, inhibiting the NF-κB activity abrogated the release of these cytokines [25]. Starting from the initial studies carried out on autoantibodies, considerable progress has been made in identifying other possible molecular mechanisms implicated in the activation of NF-κB, which could explain the chronic inflammatory situation characteristic of SS. Accumulated data suggest that the multiple roles of NF-κB in pSS could be related to the dynamic context of dysregulated inflammatory factors observed in pSS.

## 3. NF-κB Transcription Factors

The family of NF-κB is composed of five members of proteins linking to DNA, (RelA, RelB, RelC, NFκ-B1, and NFκ-B2) that trigger a set of inflammatory downstream effectors after nuclear translocation, involved in a broad range of biological processes. Either a canonical or a non-canonical pathway can be responsible for their activation; the canonical pathway mediates inflammatory responses and leads to a rapid but transient NF-κB activation, while the non-canonical signalling is a slow, long-lasting pathway (see Figure 1). Typical inducers of the non-canonical NF-κB pathway are ligands of a subset of the tumour necrosis factor receptors (TNFR) superfamily involved in the differentiation of the immune system, as well as in secondary lymphoid organogenesis [30]. The NF-κB family members show homology through a 300 amino acid N-terminal DNA binding/dimerization domain, named the Rel homology domain (RHD). The RHD is a complex system where family members can constitute homodimers and heterodimers, which are normally kept inactive in the cytoplasm through interaction with inhibitory proteins of the IκB family (IκBs) [31]. The common regulatory step in both the canonical and non-canonical cascades is the activation of an IκB kinase (IKK) complex consisting of catalytic kinase subunits (IKKα and/or IKKβ) and the regulatory non-enzymatic scaffold protein NF-κB essential modulator (NEMO), also known as IKKγ [31]. NF-κB dimers are activated by IKK-mediated phosphorylation of IκBs, which triggers proteasomal IκBs degradation, liberating the dimers of NF-κB from the NF-κB-IκBs complex, that subsequently translocate to the nucleus, linking to κB enhancer elements of target genes [32] (Figure 1). Among the IκBs, the best characterized is IκBα, which requires degradation of the activity of IKK kinase. IκBα functions as a negative feedback loop to sequester NF-κB subunits, because NF-κB activation induces the expression of the IκBα gene, which terminates signalling unless a persistent activation signal is present [32].

NF-κB is activated in every cell type and has a central role in inflammation. It is, therefore, fundamentally implicated in the molecular pathways that induce the transcription of pro-inflammatory genes [17,18,25,26,33,34]. NF-κB is highly activated in various inflammatory disorders and triggers the transcription of chemokines, cytokines, pro-inflammatory enzymes, adhesion proteins, and other factors to modulate the inflammatory response, such as metalloproteinases, Cox-2, and inducible nitric oxide synthase [35,36,37], although the mechanism is still unclear (a schematic representation of NF-κB activation is reported in Figure 2). In RA, NF-κB is highly expressed in the inflamed synovial stratum [38,39], where it enhances the engagement of inflammatory cells and pro-inflammatory cytokines production such as IL-1, IL-6, IL-8, and TNF-α [38,39]. Interestingly, recent evidence has shown that alterations in the modulation of NF-κB-dependent gene expression lead to a variety of other inflammatory and autoimmune disorders, neurological conditions and cancers [33,34]. In pSS, a correlation between NF-κB signalling and chronic inflammation has been demonstrated by various reports; nuclear translocation of NF-κB into focal infiltrated lymphocytes and into the acinar epithelium surrounding the infiltrates from SGs of patients with pSS was detected, while distal normal acini and ductal structures showed no nuclear translocation [1]. In addition, the non-canonical NF-κB p65 nuclear translocation has been induced in pSS SGEC by a range of molecular agents such as epidermal growth receptor (EGFR) and B-cell activator CD40 [40,41]. Furthermore, a downregulated gene and protein expression of IκBα was detected in pSS monocytes, contributing to an enhanced NF-κB activity. Finally, pSS autoantibodies can trigger the NF-κB signalling pathway, thus, contributing to exacerbate the inflammatory condition [13].

Based on all these assumptions, that are extensively documented, we report a close review of the recent literature on the molecular mechanisms that check NF-κB activation in pSS and on the potential of new pharmacological interventions for optimizing pSS treatment regimes.

## 4. Small-Molecule Inhibitors of NF-κB in Sjögren’s Syndrome

The modulation of the NF-κB pathways has frequently been described as “pro-inflammatory”, largely due to the key role of NF-κB in the pro-inflammatory genes expression including cytokines, chemokines, and adhesion molecules [42,43]. Many findings demonstrated that epithelial cells in the glandular sites of patients affected by pSS are able to release factors that address the chemoattraction of lymphocytes and promote chronic inflammatory responses [12,15,16,24,25,26]. NF-κB pathway modulation was therefore investigated in pSS, highlighting a role in regulating the production of pro-inflammatory cytokines, leukocyte enrolment, or cell survival [17,18,25,26,44]. In pSS, the NF-κB activation cascade can be modulated at different levels [30]. Considering the correlation with the biopsy focus score, grade of infiltration and evaluated disease activity, phosphorylated IKKε, responsible for the degradation of IkB proteins, were significantly and positively correlated with NF-κB levels in pSS [4]. The levels of B-Cell Activating Factor (BAFF) and those of numerous pro-inflammatory cytokines, all regulated by NF-κB signalling, are augmented in pSS [45]. Nucleotide polymorphisms in NF-κB pathway genes have been linked with pSS [46], and a specific mutation in the Ikα-826T, one of the promoters of a member of the inhibitory IkB complex, was associated with susceptibility of pSS [47,48]. Numerous small molecule inhibitors of the NF-κB signalling pathways are currently commercially available for use, and NF-κB modulators are under study in clinical trials for pSS treatment [19,25,30,49,50,51,52,53,54]. Many preclinical studies have already analysed the role of NF-κB signalling in the glandular tissue in pSS. The pSS SGECs have been recognized to have an active NF-κB pathway. The phosphorylated forms of IKKε, IκBα, and NF-κB were expressed in the ductal cells in minor SGs derived from pSS patients [19]. By stimulating the Toll-Like Receptor 2 (TLR2) in SGECs, IL-2 production was induced through the NF-κB cascade in pSS SGECs [49,50,51]. SGECs treated with the anti-Ro/SSA autoantibodies isolated from pSS patients showed a progressive increase in constitutive NF-κB activation, and transfection of SGECs with IκBα in SGECs treated with anti-Ro/SSA led to a remarkable production of pro-inflammatory cytokines and an enhanced apoptosis [25]. Furthermore, recent findings showed that gene silencing of the natural NF-κB inhibitor TNF Alpha Induced Protein 3 (TNFAIP3) in keratin-14-positive epithelial cells, promoting the activation of the constitutive NF-κB cascade, induces the initial phases of pSS, leading to a reduced production of saliva and lymphocyte invasion in the SGs [52]. This effect is likely related to the calcium pathway in the acinar cells, since calcium signalling has an important role in NF-κB pathway activity [53,54]. A list of NF-κB small molecule inhibitors tested in pSS is reported in Table 1.

### The Key Role of IκBα in NF-κB Modulation in pSS

Among the well-characterized regulators of NF-κB activation in SGEC, IκBα is particularly important for the pathogenesis of pSS. The concept that IκBα expression negatively regulates NF-κB DNA binding activity was demonstrated by the fact that reduced IκBα overlaps with nuclear translocation of the NF-κB and the appearance of NF-κB activity [55]. For SGs, adenoid cystic carcinoma of human SGs cell lines, stably transfected with the mutant IκBα expression vector (IκBαM) share an effectively cancelled constitutive and liposaccharide-induced NF-κB activity, concomitantly with a significantly diminished VEGF gene and protein expression. This effect leads to a lower endothelial cell mobility and, thus, might represent a promising anti-angiogenesis strategy in adenoid cystic carcinoma (ACC) therapy [56]. In pSS SGEC, abnormal levels of IκBα were detected in comparison with those in healthy subjects, showing a clear reduction of IκBα in salivary tissues from active pSS patients [19]. This was confirmed in biopsy specimens, where a moderate IκBα positive staining located in the cytoplasm of acini and ductal cells was revealed in healthy controls, whereas in pSS salivary gland biopsies the cytoplasmic positivity for IκBα was very weak [19]. All of this suggests that the production of proinflammatory cytokines occurs through the persistent activation of NF-κB signalling [17,18,25,26]. In addition, a reduced gene and protein expression of IκBα was demonstrated in monocytes from pSS patients in comparison with healthy subjects, suggesting that the reduced expression of this NF-κB inhibitor may reflect an increased inflammatory response [26]. Specifically, published data show that mutations in IκBα are linked to inflammatory autoimmune disorders. An 8-bp insertion in the promoter region of IκBα represents a protective factor against the development of primary progressive multiple sclerosis [57]. Klein et al. showed that IκBα polymorphisms might also be associated with Crohn’s disease [58], SLE [59] and pSS [47,48]. In particular, mutant mice, that have defective IκBα expression, showed a shorter lifetime, hypersensitivity to septic shock and altered T cell development, all features of pSS [47,48]. Furthermore, overexpression of the NF-κB repressor, IκBα, determines an inhibitory effect on the production of STAT-4 protein, a transcription factor activated by interleukin 12 whose gene polymorphism was recently linked to pSS [60,61].

NF-κB signalling activation and termination is secured by various regulatory processes. In view of the well-characterized links between NF-κB and pSS disease, disentangling the complexity of NF-κB modulation is an essential goal in order to find effective, more specific therapeutic agents for the treatment of pSS.

## 5. Impaired NF-κB Signalling Activated by EDA-A1/EDAR in pSS Salivary Glands

In addition to its role in mediation inflammation, NF-κB is also essential for developing the epidermal derivatives, hair, nails, and SGs [62]; a series of molecular signals is now well defined, beginning with the binding of ectodysplasin (EDA-A1) to the EDA-receptor (EDAR), components of the tumour necrosis factor α (TNFα)-related signalling pathway [62]. EDA-A1 signalling is recognized as an important evolutionarily conserved pathway regulating the formation and patterning of vertebrate skin appendages, including SGs [63]. When these genes show mutations, a condition known as hypohidrotic (or anhidrotic) ectodermal dysplasia (HED/EDA) occurs [64]. The NF-κB pathway that mainly impinges on EDA-A1/EDAR-dependent SGs branching morphogenesis is the canonical NF-κB activation cascade [65].

Over the last years, great progress has been made in identifying the key molecular regulators controlling NF-κB activation, and a repertoire of crucial self-regulators ensuring the termination of NF-κB responses has been identified [66]. Interestingly, this well-orchestrated biological process may undergo alterations [33,34,67] and, consequently, deregulated NF-κB activation contributes to the autoimmune diseases pathogenesis, characterized by an intense inflammatory response [34,67]. As a matter of fact, NF-κB was demonstrated to play a salient role in the pathological development of pSS, correlated with the intense chronic inflammation findings in this disease [17,18,25,26,48]. In this context, several studies conducted on SGEC derived from pSS patients investigated the mechanism-of-action of the NF-κB cascade and performed target identification in the deregulated inflammatory situation. Recent findings demonstrated that EDA-A1 induces several genes involved in the synthesis of the NF-κB pathway molecules, including the feedback inhibitors IκBα and TNFAIP3. IκBα is known to be expressed in hair placodes and SGs [68], and TNFAIP3, a key negative feedback regulator of the NF-κB signalling cascade, plays a role in the EDA-A1, EDAR and EDAR-associated death domain (EDARADD) genes control, which results mutated in HED/EDA [69]. Therefore, recently, chemokines have been revealed as immediate target genes of the EDA/NF-κB pathway, leading to modulation of the multiple signalling pathways implicated in skin appendage development; when this scheme is deregulated, an inflammatory process may be induced [69]. Against this background, a recent study has investigated the EDA-A1 and EDAR genes and proteins expression in pSS SGs, showing that TNFAIP3 is deregulated in pSS SGEC. This results in an increased and excessive EDA-A1/EDAR gene and protein expression in pSS SGEC that determines a correlated high induction of NF-κB [70] (Figure 3). Furthermore, TNFAIP3 gene knockdown performed on healthy SGEC, through the application of the siRNA gene silencing technology, determined an over-activation of the EDA-A1/EDAR expression and consequently NF-κB nuclear translocation and activation [70] (Figure 3). The authors have shown that, in pSS SGEC, NF-κB is activated downstream of EDA-A1/EDAR signalling and after transfecting pSS SGEC with the mutated form of the regulatory protein IκBα, the EDA-A1/EDAR-NF-κB signalling pathway was affected in SGs, suggesting that the IκBα-dependent canonical NF-κB cascade was active in pSS SGEC [70]. This recent discovery suggests that the pathways involved in ectodermal development and inflammation may be fundamentally the same, but lead to target gene activation depending on the cell type and/or on the specific pathological condition features. The implication of the NF-κB pathway in development was a very surprising finding, because it is involved primarily in TNF-α receptors-mediated inflammation and immunity; now, the recurrent question is how cells can distinguish between the NF-κB pathway activation signals, as well as how specific target genes activation is precisely and independently controlled during developmental or inflammatory events.

## 6. Toll-Like Receptor-Mediated NF-κB Activation in pSS

A large volume of recent evidence underlines the finding that the SGs epithelium is the major actor in the promotion and progression of the chronic inflammatory reactions observed in pSS, through the induction of pro-inflammatory cytokines and chemokines [8,71]. The innate immune system uses a diverse set of recognition receptors to activate the intracellular signalling pathway, such as Toll-like receptors (TLRs) molecules. Indeed, TLRs activation lead to the recruitment of adaptor proteins within the cytosol, that culminates in signal transduction resulting in the transcription of genes involved in chronic inflammation [72,73]. TLRs were initially identified as receptors important only in host defences, but it is now clear that the TLRs, for example TLR2 and TLR4, are crucial in autoimmunity development [74,75,76], as demonstrated in RA [74], SLE [77], multiple sclerosis [78], and inflammatory bowel diseases [79]. Studies comparing mice and humans revealed that numerous types of epithelial cells express TLRs, supporting the hypothesis that the epithelium represents the first line of defence of the innate immune system [80,81]. These observations were confirmed also in pSS; the induction of TLRs signalling in SGEC leads to the release of inflammatory mediators, including IL-6, IL-8, and TNF-α [72], which are critical mediators of the inflammatory processes of pSS. In addition, recent works have evidenced the important contribution of TLRs activation to the initiation and progression of the pSS pathogenesis. In particular, TLR2, TLR3, and TLR4 are expressed on the SGEC membrane [82], and in addition, immunohistochemical analyses of TLR2, 3 and 4 on labial SG tissue from pSS patients confirmed a significantly higher constitutive expression of these receptors, found in SG-infiltrating mononuclear cells as well as acinar cells and ductal SGEC, supporting the intrinsic epithelial activation in pSS [40]. TLR4, in particular, resulted highly expressed specifically in infiltrating mononuclear cells and in ductal and acinar cells [83,84] of pSS SGs, and receptor levels were correlated with the degree of glandular inflammation [83,84]. At the same time, investigations conducted on pSS peripheral blood mononuclear cell (PBMC) confirmed a dysregulation of TLR7, 8 and 9 molecules compared to controls, where TLR7 and 8 recognize single-stranded RNA [85], while TLR9 is activated by un-methylated CpG DNA [86]. This led to an altered recognition of DNA and RNA, eventually resulting in the development of pSS. [87].

### Role of TLRs in the NF-κB-Mediated Inflammatory State in pSS

Several authors now agree that TLRs trigger an intracellular cascade of molecular events, which has, as its final step, NF-κB activation. Active NF-κB determines the transcription of inflammatory cytokine genes responsible for the exacerbation of inflammation [73]. Studies conducted on experimental animal models confirm that an intensely inflammatory microenvironment could be the basis of autoimmune diseases [88]. This scenario seems to be plausible also for pSS. A recent study reported that TLR-7 and its downstream signalling factors are strongly expressed in labial SG of pSS patients. The authors observed that TLR-7 downstream molecules are expressed in pSS SGEC after TLR-7 ligand stimulation in vitro, inducing the activation of the NF-κB pathway, which elicits the release of inflammatory factors such as IFN-α and IFN-γ [89]. Furthermore, Kwok et al. demonstrated an increased expression of TLR2, TLR4, and TLR6 in pSS SGs, in association with IL-17, IL-6, and IL-23 over-expression, factors that promote T helper17 (Th17) differentiation and amplification. The signalling pathway starts with TLR2 stimulation, which induces a cascade that involves the activation of TLR4 and TLR6. This determines the production of IL-17 and IL-23, which, as demonstrated by the authors, occurs through IκBα phosphorylation, the IL-6, signal transducer and activator of transcription 3 (STAT3), and NF-κB pathways [90]. However, the ligands that eventually activate TLR2 in the context of pSS are still doubtful and little known. Using peptidoglycan (PNG) as stimulus for TLR2 activation, an increased expression of immune mediators (ICAM-1, CD40, and MHC-1) was observed in SGECs derived from pSS patients and controls [82]. In a corroborative study using SGEC from pSS patients, TLR2 drove the NF-κB-dependent secretion of IL-15 [50,51] as confirmed using antibodies anti-TLR2 to block IL-15 secretion [50,51]. Furthermore, by using the dominant-negative inhibitory IκBα vector to inhibit NF-κB activation, TLR2-dependent IL-15 production was reduced, suggesting a transcriptional level control [50,51]. Therefore, this study underlines the importance of the TLR2/IL-15/ NF-κB pathway as a strong potential candidate for the therapeutic modulation of pSS, ameliorating both local and systemic pSS disease manifestations [50,51] A schematic representation of the TLR2 molecular pathway activation in pSS is reported in Figure 4.

## 7. Modulation of NF-κB Activation by the Anti CD-20 Monoclonal Antibody Rituximab

The management of pSS patients is essentially symptomatic, no curative agents for SS yet exist and demonstrations of the efficacy of systemic drugs are lacking. Given the key role of chronic B-cell activation in pSS, B-cell target therapies based on B-cell downregulation have been individuated as the first potential candidates. CD20’s attractiveness as a therapeutic target derived from the growing understanding of the molecular basis for several properties related to its structure and its interaction networks [91]. CD20 is a non-glycosylated surface phosphoprotein, found on a variety of healthy and malignant B cells, whose function is probably involved in calcium influx [92,93]. CD20 expression appears early during B cell maturation but is lost during B-cell differentiation into plasma cells [94]. For many years, the function of CD20 in normal immune physiology remained poorly defined, based on few data demonstrating a role in the generation of the long-term humoral response [95].

### Hypothetical Scenario Involving RTX as a Negative Regulator of the NF-κB Pathway in pSS

Rituximab (RTX), a mouse/human chimeric monoclonal antibody directed against CD20 antigen on B cells surface, represents a treatment for both pSS and SS-related malignant lymphoproliferative disease [96], whose efficacy has been investigated in the last decade, in the presence (or not) of a lymphoproliferative disorder [97,98,99], owing to its proven efficacy in other chronic inflammatory diseases, such as RA [100] and systemic vasculitis [101].

Recent experimental evidence demonstrated that in a co-culture system of pSS SGEC with pSS lymphocytes, RTX stimulation causes B cells depletion, leading to a drastically reduced transcription of pro-inflammatory mediator genes and protein secretion. This report suggests that B-lymphocytes regulate the cytokine and chemokine release by pSS SGEC because of their proximity in inflammatory areas. This intrinsic activation of SGEC exacerbates the inflammation, further modulating the release of inflammatory factors along post-translational pathways [102,103]. In this hypothetical scenario, a decisive role could be played by the inhibition of the constitutive activation of NF-κB. Treatment with RTX of pSS SGEC co-cultured with pSS B-lymphocytes, determines a lower NF-κB DNA binding activity in the SGEC, so inhibiting the pro-inflammatory genes transcription [102,103]. Now, RTX interferes with the constitutive activation of the NF-κB pathway through the modulation of Raf-1 kinase inhibitor protein (RKIP) expression [104], which acts directly by down-regulating IkB kinase (IKK) activity and indirectly by interfering with IKK activators [105]. RKIP is believed to play an important role in various inflammatory diseases and cancers [106] and results constitutively under-expressed in pSS SGEC [102]. These data suggest that RKIP could increase NF-κB activity, leading to the persistent chronic inflammatory condition characteristic of pSS. Therefore, the function of RTX as a negative regulator of the NF-κB pathway in pSS SGEC is based on the modulation of RKIP expression; RTX, in fact, up-regulates RKIP expression in pSS SGEC, and RTX-mediated RKIP induction diminishes the phosphorylation of the components of the NF-κB pathway [102] (Figure 5). Experimental RKIP gene silencing in pSS SGEC confirmed this hypothesis, leading to pro-inflammatory cytokine secretion by pSS SGEC, and preventing NF-κB inhibition. However, what is the effect of RTX on NF-κB relate to the B cells depletion, since literature indicates that treatment with RTX leads to an effective depletion of B cells in pSS patients? Evidence suggests that Fc/FcγR interactions are critical, as determined in both animal models and humans [107]. Data collected suggest that the formation of IgG immune complexes between B lymphocytes and RTX could engage specific FcγR on pSS SGEC, resulting in a decreased NF-κB activity and interruption of the NF-κB signalling pathway through the up-regulation of the RKIP protein [102] and engagement of the mitogen-activated protein kinase (MAP kinase) signalling [108]. A schematic model of RTX-mediated inhibition of the NF-κB pathway in pSS SGEC is reported in Figure 5.

## 8. Fine Modulation of NF-κB Activity by TNFAIP3

Numerous studies reported in this review clearly show that several cell types isolated from patients affected by autoimmune diseases show constitutively activated NF-κB transcription factors; there is considerable evidence of NF-κB activation in SGECs derived from pSS patients [17,18,25,26]. Dysregulation of NF-κB-dependent gene expression leads to a variety of autoimmune inflammatory conditions, cancer and neurological disorders [33,34]. Since NF-κB signalling activation is important for several cellular processes, not surprisingly, a tight modulation of this pathway is absolutely essential to trigger target genes. As reported above, among the small regulators of NF-κB activity, great attention has been paid, in the last years, to TNFAIP3, which is a negative feedback regulator of NF-κB activation via TNF-α signalling. Given its key role in the fine modulation of NF-κB pathway, it has been demonstrated that a dysregulated expression of TNFAIP3 protein contributes to chronic inflammation and tissue injury [109]. The importance of TNFAIP3 in reducing inflammation is underlined by the linking of TNFAIP3 genomic region polymorphisms with human autoimmune and inflammatory diseases, including RA [110], psoriasis [111], SLE [112], and type 1 diabetes [113]. Thus, TNFAIP3 has been considered as a crucial anti-inflammatory factor acting to limit prolonged inflammation. A presumed association of TNFAIP3 polymorphism with pSS syndrome has recently been reported [114]. Moreover, TNFAIP3 gene and protein expression levels resulted diminished in salivary tissue from active pSS, demonstrating that under-expression of this protein may reflect an enhanced inflammatory reaction.

### Reduced TNFAIP3 Expression Levels in pSS Affect NF-κB Signalling

Recent investigations support an anti-inflammatory role of TNFAIP3, indeed, knockout mice for this gene evolve multiple organ inflammation [115], TNFAIP3 gene silencing in dendritic cells leads to the release of specific co-stimulatory factors, such as pro-inflammatory cytokines [116] and genetically TNFAIP3 deficient mice also show severe intestinal inflammation [115]. The reduced levels of TNFAIP3 observed in pSS, characterized by a remarkable inflammation of the SGs, may promote chronic invasive immune processes in these patients, triggering an initial abnormal inflammatory response. Several findings suggest that the reduction of TNFAIP3 expression levels could lead to the deregulation of NF-κB signalling in pSS patients who show a higher transcriptional activity of NF-κB than normal control subjects [17]. Since TNFAIP3 is a negative regulator of NF-κB signalling in human SGECs, its deregulation could be responsible for the persistent expression of NF-κB that occurs in pSS. This corroborates the notion that human SGECs play an essential role in coordinating the SGs inflammatory reactions to pro-inflammatory factors and suggests that the NF-κB pathway is crucial in these cells for modulating immune responses (see, for example, Figure 3) [17]. Since TNF-α contributes to tissue inflammation, a system mediated by TNF-α-dependent NF-κB activation, it is plausible to postulate that it may translocate the nucleus and promote the expression of inflammatory genes [117]. In accordance with this hypothesis, our recent findings have shown a higher expression of TNF-α in SGECs isolated from pSS patients [15], confirming the role of TNF-α as an inducer of TNFAIP3 protein expression. The enhanced NF-κB activity that occurs in human SGECs, following treatment of the epithelium with TNF-α, could be responsible for the paracrine progression of the inflammatory response shown in SS, inducing pro-inflammatory genes transcription. Therefore, because TNFAIP3 is affected in the negative regulation of NF-κB activation, the inactivity of TNFAIP3 protein was postulated to give rise to a constitutive activation of NF-κB contributing to marked inflammatory reactions. These hypotheses were demonstrated in TNFAIP3 knockdown experiments showing that TNFAIP3 gene silencing induces a constitutive activation of NF-κB in human healthy SGEC [17] (Figure 6) and in experiments conducted on TNFAIP3 knockout mice [115]. Mice with deficient TNFAIP3 are, in fact, hypersensitive to TNF-α and showed grave inflammation and severe damage in multiple organs. TNFAIP3-deficient cells are not able to terminate TNF-α-induced NF-κB responses and rapidly die due to TNF-α-mediated apoptosis [115].

## 9. Conclusions

In this review, we summarize the aberrant activation of NF-κB in pSS, clearly demonstrating that NF-κB has a crucial role in the pathogenesis of pSS, since it promotes chronic inflammation. NF-κB, through intrinsic SGEC activation, regulates sophisticated feedback circuits in pSS that comprise all elements of the cellular immune response. Since NF-κB and members of its signalling pathways regulate cellular activity from DNA transcription to translation into proteins, efficient and properly controlled NF-κB signalling is important during physiological immune homeostasis. In fact, the integrity of the signal triggered by NF-κB is essential in preventing the onset of pSS autoimmune disease. It is noteworthy that the significance of NF-κB activation in pSS suggests that inhibition of this signalling pathway could provide novel strategies for the prevention and treatment of the SGs dysfunction characterizing pSS. New future perspectives suggest, for example, the use of IKKε inhibitors for the treatment of pSS, repressing downstream NF-κB signalling activation. Furthermore, great attention is now being paid to the modulation of NF-κB activity and expression of NF-κB target genes through an IκBα-mediated negative feedback mechanism. Hopefully, as our understanding of the regulation of the NF-κB pathways increases, insights into a better design of drugs that can effectively target NF-κB for the prevention and treatment of pSS may be gained.

**Author contributions:** All authors conceived of the presented idea and approved the final version of the manuscript. M.S. and S.L. collected the data reported in the study and take responsibility for their integrity. D.R. performed a critical reading of this review. All authors have read and agreed to the published version of the manuscript.

## Figures and Tables

**Figure 1 jcm-09-02821-f001:**
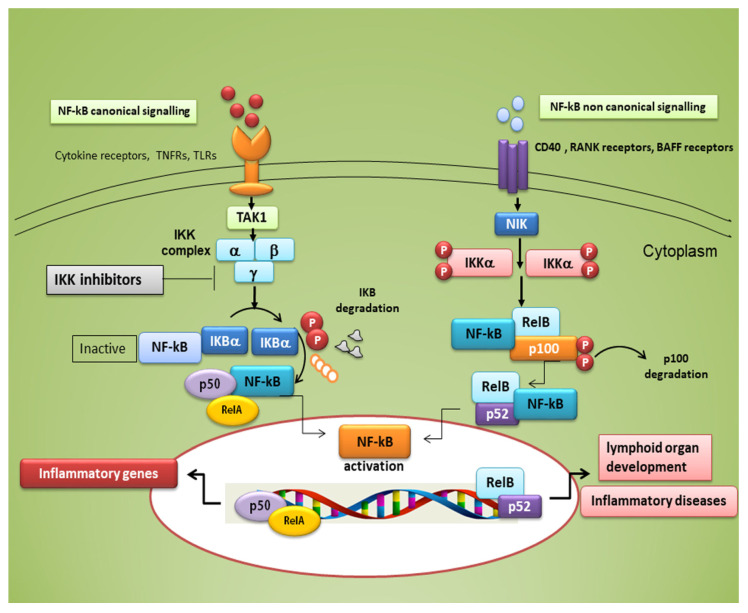
Canonical and non-canonical nuclear factor κ (kappa)-light-chain-enhancer of activated B cells (NF-κB) signalling pathways. The canonical NF-κB pathway starts with the activation of innate and adaptive immune receptors in response to various ligand molecules, which transfers the signal across the cell membrane causing the activation of the trimeric IκB kinase (IKK) complex, composed of catalytic (IKKα and IKKβ) and regulatory (IKKγ) subunits. The IKK complex phosphorylates IκBα and phosphorylated IκB undergo ubiquitylation and proteasomal degradation, allowing nuclear translocation of the RelA/p50 dimer of the NF-κB heterodimer. The non-canonical NF-κB pathway selectively responds to a subset of TNFR members that induce the activation of the NF-κB-inducing kinase (NIK). NIK phosphorylates and activates IKKα, which in turn phosphorylates carboxy-terminal serine residues of p100, triggering selective degradation of the C-terminal IκB-like structure of p100, and mediates the persistent activation of the RelB/p52 complex.

**Figure 2 jcm-09-02821-f002:**
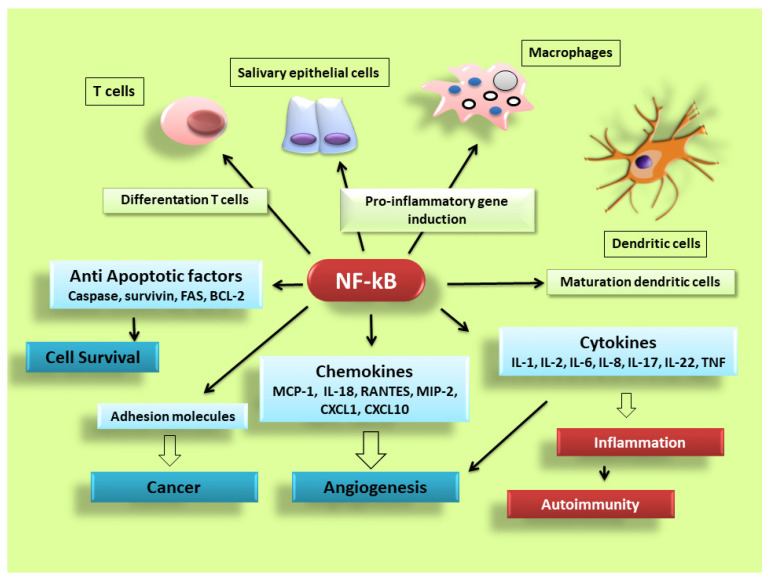
NF-κB signalling pathway dictates the inflammatory responses. After its activation, it can induce the transcription of a large number of genes including pro-inflammatory cytokines, chemokines, adhesion molecules, cell cycle regulatory molecules, anti-apoptotic proteins and angiogenic factors, and thereby regulate cell proliferation, apoptosis, morphogenesis, differentiation, angiogenesis, and inflammation. The regulation of inflammation, cell proliferation and apoptosis is central to the understanding of many diseases, such as autoimmune diseases and cancer.

**Figure 3 jcm-09-02821-f003:**
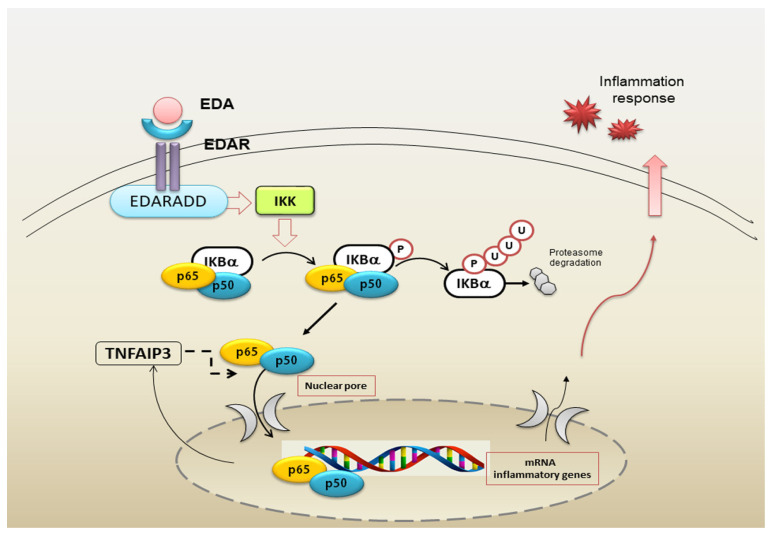
Schematic overview of the EDA/EDAR/canonical NF-κB pathway. The EDA isoform of the TNF-α family member Ectodysplasin interacts with its receptor EDAR leading to the recruitment of EDARADD (death domain adaptor); in turn, this complex activates the IKK complex. The IKK complex phosphorylates IκBα, that undergoes ubiquitylation and proteasomal degradation, inducing nuclear translocation of the NF-κB heterodimer RelA/p50 that triggers the transcription of pro-inflammatory genes, including those that encode the negative regulators IκBα and TNF-α-induced protein 3 (TNFAIP3). EDA: Ectodysplasin-A; EDAR: Ectodysplasin-A Receptor; EDARADD: Ectodysplasin-A receptor-associated associated death domain.

**Figure 4 jcm-09-02821-f004:**
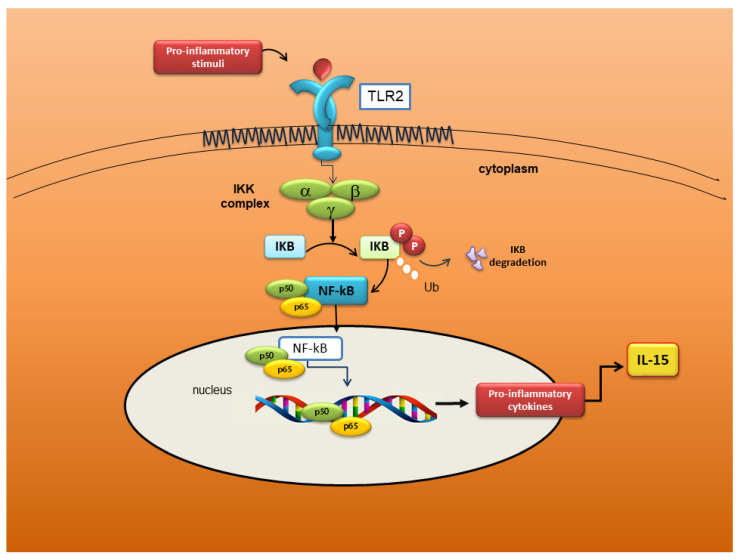
Schematic representation of the Toll-Like Receptor 2 (TLR2)/IL-15/NF-κB pathway in pSS SGEC. TLR2, in response to pro-inflammatory stimuli, activate the NF-κB pathway; in the nucleus, the active NF-κB promote IL-15 gene transcription; thus, incrementing inflammatory disorders in SGs of pSS patients.

**Figure 5 jcm-09-02821-f005:**
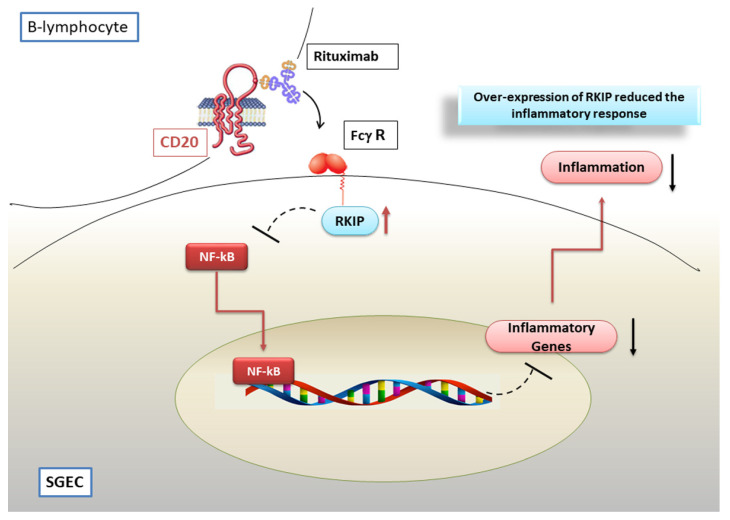
Rituximab (RTX) inhibits NF-κB signalling in pSS SGEC. The pSS SGs epithelial cells (SGEC) were found to express low levels of Raf-1 kinase inhibitor protein (RKIP). In a co-culture system with pSS B-lymphocytes, the FcγR-mediated interaction of RTX/CD20 induces the upregulation of RKIP expression, decreasing NF-κB activity, and, consequently, inhibiting the pro-inflammatory genes transcription.

**Figure 6 jcm-09-02821-f006:**
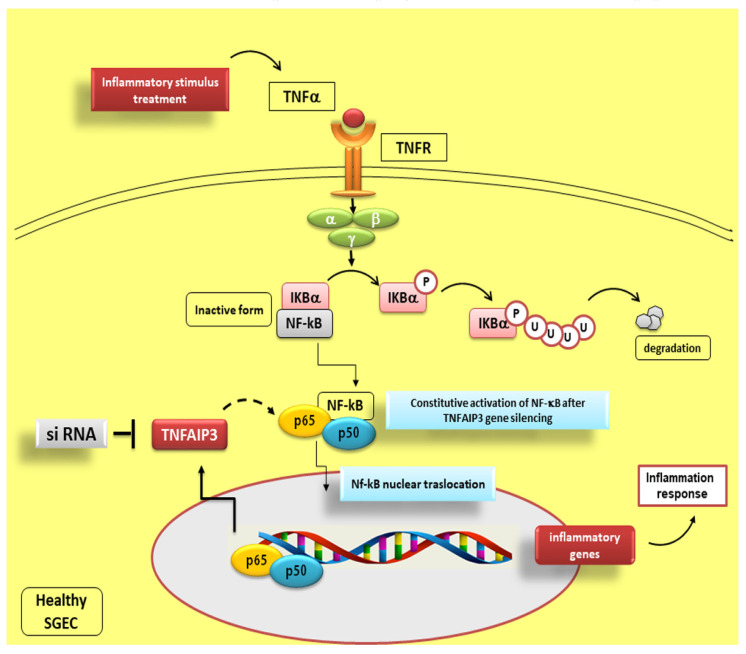
Effect of TNFAIP3 gene silencing in TNF-α stimulated healthy SGEC. This scheme shows how TNFAIP3 knockdown experiments induce a constitutive activation of NF-κB in human healthy SGEC, leading to a severe inflammatory response. siRNA: short interfering RNA; dashed line represents inhibition of the constitutive activation of NF-κB; solid line indicates activation.

**Table 1 jcm-09-02821-t001:** List of NF-κB small molecules inhibitors tested or identified in primary SS (pSS).

Small Inhibitors of NF-kB in pSS	Study	References
**Iguratimod**	Clinical study	[30]
**Syk-inihibitor-Gs-9876**	Clinical study	[30]
**IKKε**	Pre-clinical study	[4,30]
**IκBα**	Pre-clinical study	[19,26,47,48]
**anti-Ro/SSA autoantibodies**	Pre-clinical study	[14,16,24,25]
**TNFAIP3**	Pre-clinical study	[17]
**Calcium mobilization**	Pre-clinical study	[53,54]

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
