# Peer review of "Understanding the Complexity of Sjögren’s Syndrome: Remarkable Progress in Elucidating NF-κB Mechanisms"

_jcm, 2020, doi:10.3390/jcm9092821_

Round 1

Reviewer 1 Report

In this manuscript, the authors conducted a thorough review of the role of NF-KB-related mechanisms in the pathogenesis and potential treatments of Sjögren's syndrome. This is a comprehensive review that summarises current evidence in this topic in a structured fashion. A few minor points should be addressed, though, to improve clarity.

  • An English language spelling/grammar check should be completed. Some sentences/paragraphs would improve shortening/reformulation to enhance readability.
  • Figure 2- please consider adjusting TNF alpha to TNF
  • Section 4, Lines 168-199 - please address if it is possible to convert this paragraph into at least 2 different paragraphs. The information is relevant but can be slightly improved in form for better reading.
  • Section 4, Lines 168-199 - the title mentions small molecule inhibitors but none are cited in the text (lines 183-185 reference to another article). Please list eg in table which available NFKB inhibitors have been tested in pSS (clinical and preclinical).
  • Line 210 - add reference.
  • Line 264 - study instead of finding
  • Figure 3 - define EDA/EDAR in legend.
  • Section 6, Lines 294-348 - please address if it is possible to convert this paragraph into at least 2 different paragraphs. The information is relevant but can be slightly improved in form for better reading.
  • Section 7, Lines 357-403 - please address if it is possible to convert this paragraph into at least 2 different paragraphs. The information is relevant but can be slightly improved in form for better reading.
  • Line 414-415 - please attenuate this statement.
  • Section 8, Lines 412-462 - please address if it is possible to convert this paragraph into at least 2 different paragraphs. The information is relevant but can be slightly improved in form for better reading.
  • Figure 6 - please define dashed line (vs solid line); also define si RNA.

Author Response

We would like to express our sincere gratitude to the reviewers for their constructive and positive comments and for the very thoughtful critique of our manuscript and are pleased to say that we tried to address all the concerns raised. All changes to the manuscript are indicated in the text by highlighting.

We respond below in detail to each of the reviewer’s comments. We hope that the reviewers will find our responses to their comments satisfactory, and we are willing to finish the revised version of the manuscript including any further suggestion that the reviewers may have.

Reviewer 1:

The authors sincerely appreciate the reviewer’s comments which allows to improve the manuscript.

An English language spelling/grammar check should be completed. Some sentences/paragraphs would improve shortening/reformulation to enhance readability.

The revised manuscript was carefully checked and corrected by a native English speaker as you suggested.

Figure 2- please consider adjusting TNF alpha to TNF

We changed TNF alpha to TNF in Figure 2.

Section 4, Lines 168-199 - please address if it is possible to convert this paragraph into at least 2 different paragraphs. The information is relevant but can be slightly improved in form for better reading.

We agree with this suggestion proposed by the reviewer and we divided the paragraph 4 titled “Small-molecule inhibitors of NF-kB in Sjögren’s syndrome” into 2 paragraphs.

Section 4, Lines 168-199 - the title mentions small molecule inhibitors but none are cited in the text (lines 183-185 reference to another article). Please list eg in table which available NFKB inhibitors have been tested in pSS (clinical and preclinical)

As you requested, we have created a table in which we have summarized the molecules currently tested as NF-kB inhibitors in the clinical or preclinical phase

Line 210 - add reference.

We added the missing reference.

Line 264 - study instead of finding

We done, as suggest.

Figure 3 - define EDA/EDAR in legend.

We defined EDA/EDAR/EDARADD in the legend.

Section 6, Lines 294-348 - please address if it is possible to convert this paragraph into at least 2 different paragraphs. The information is relevant but can be slightly improved in form for better reading.

As you requested, section 6 has been divided into two subparagraphs.

Section 7, Lines 357-403 - please address if it is possible to convert this paragraph into at least 2 different paragraphs. The information is relevant but can be slightly improved in form for better reading.

We agree with this suggestion proposed by reviewer and we divided the paragraph 7 titled “Modulation of NF-kB activation by the anti CD-20 monoclonal antibody Rituximab” into 2 paragraphs.

Line 414-415 - please attenuate this statement.

As rightly suggest, we attenuated the sentence.

Section 8, Lines 412-462 - please address if it is possible to convert this paragraph into at least 2 different paragraphs. The information is relevant but can be slightly improved in form for better reading.

We agree with this suggestion proposed by reviewer and we divided the paragraph 8 into 2 paragraphs.

Figure 6 - please define dashed line (vs solid line); also define si RNA.

We defined dashed line in the Legend of the Figure 6 and also clarified siRNA in the same figure.

Reviewer 2 Report

Thanks for the opportunity to read this paper. I commend the authors on tackling an important complex topic and presenting the review about the role of NF-kappa B in the pathogenesis of the primary Sjögren's syndrome. The manuscript is concise and clearly written. 

We have no effective therapies in primary Sjögrens syndrome (pSS)A better understanding of the mechanisms underlying the pathological activation of NF-κB in pSS is crucial for designing more specific and effective therapeutic agents for the treatment of the pSS.

Could the authors add future perspectives of understanding of the mechanisms underlying the pathological aberrant activation of NF-κB in pSS?

Author Response

We would like to express our sincere gratitude to the reviewers for their constructive and positive comments and for the very thoughtful critique of our manuscript and are pleased to say that we tried to address all the concerns raised. All changes to the manuscript are indicated in the text by highlighting.

We respond below in detail to each of the reviewer’s comments. We hope that the reviewers will find our responses to their comments satisfactory, and we are willing to finish the revised version of the manuscript including any further suggestion that the reviewers may have.

Reviewer 2

We thank the reviewer for careful and thorough reading of this manuscript and for the thoughtful comments and constructive suggestion, which help to improve the quality of this manuscript.

Could the authors add future perspectives of understanding of the mechanisms underlying the pathological aberrant activation of NF-κB in pSS?

The conclusions have been integrated as you requested.

Reviewer 3 Report

  This is a timely and comprehensive review focused on the role of NF-kB pathway dysregulation in the pathogenesis of Sjogren’s syndrome. The topics is crucially important in the field.   Major concerns: - Figure 2 appears poorly related to the text (lines 110-115) - The definition of Rituximab as “a new putative treatment for pSS” (line 369) appears inadequate (as demonstrated by the date of references documenting the use of Rituximab inpSS patients) - The sentence “Treatment of pSS SGECs with Rituximab determines lower NF-kB DNA binding activity in SGECs…” (lines 380-382) should be modified because, as reported in the following text and Figure 5, the effect of Rituximab on SGECs is obviously mediated by interaction with B cells - In Paragraph 8 an introduction and comment on the use of Adalimumab inpSS patients is advisable   Minor concerns: - Figure 4: TLRs in the figure should be substituted by TLR2 - the explanation of some acronyms is missing (i.e. BAFF, ACC,..) - several typos are spread throughout the text, and English editing is recommended

Author Response

We would like to express our sincere gratitude to the reviewers for their constructive and positive comments and for the very thoughtful critique of our manuscript and are pleased to say that we tried to address all the concerns raised. All changes to the manuscript are indicated in the text by highlighting.

We respond below in detail to each of the reviewer’s comments. We hope that the reviewers will find our responses to their comments satisfactory, and we are willing to finish the revised version of the manuscript including any further suggestion that the reviewers may have.

Reviewer 3:

We thank the Referee for showed interest in our work and for helpful comments that will greatly improve the manuscript and we have tried to do our best to respond to the points raised.

Figure 2 appears poorly related to the text (lines 110-115).

We moved the Figure 2 in the correct position.

The definition of Rituximab as “a new putative treatment for pSS” (line 369) appears inadequate (as demonstrated by the date of references documenting the use of Rituximab in pSS patients).

We changed the sentence according to your suggestion.

The sentence “Treatment of pSS SGECs with Rituximab determines lower NF-kB DNA binding activity in SGECs…” (lines 380-382) should be modified because, as reported in the following text and Figure 5, the effect of Rituximab on SGECs is obviously mediated by interaction with B cells.

As rightly suggest by reviewer, we improved the sentence.

In Paragraph 8 an introduction and comment on the use of Adalimumab in pSS patients is advisable.

Thank you for this suggestion, but, currently, studies on the correlation between Adalimumab and TNFAIP3 modulation are not reported in the literature. We reserve the right, in the near future, to investigate this topic and carry out experiments in this field.

Minor concerns:

Figure 4: TLRs in the figure should be substituted by TLR2

As suggest, we substituted in the figure 4, TLRs by TLR2.

The explanation of some acronyms is missing (i.e. BAFF, ACC,..)

We defined the acronyms in the text.

Several typos are spread throughout the text, and English editing is recommended

The revised manuscript was carefully checked and corrected by a native English speaker as you suggested.

This manuscript is a resubmission of an earlier submission. The following is a list of the peer review reports and author responses from that submission.